# ATG4 Mediated *Psm* ES4326*/AvrRpt2*-Induced Autophagy Dependent on Salicylic Acid in *Arabidopsis Thaliana*

**DOI:** 10.3390/ijms21145147

**Published:** 2020-07-21

**Authors:** Wenjun Gong, Bingcong Li, Baihong Zhang, Wenli Chen

**Affiliations:** 1MOE Key Laboratory of Laser Life Science & Institute of Laser Life Science, College of Biophotonics, South China Normal University, Guangzhou 510631, China; gongwenj@scnu.edu.cn (W.G.); bingcongli2018@163.com (B.L.); zhangbyhome@126.com (B.Z.); 2Guangdong Provincial Key Laboratory of Laser Life Science, College of Biophotonics, South China Normal University, Guangzhou 510631, China

**Keywords:** autophagy, enhanced disease susceptibility (EDS1), salicylic acid (SA), nonexpressor of PR genes 1 (NPR1)

## Abstract

*Psm* ES4326/*AvrRpt2* (*AvrRpt2*) was widely used as the reaction system of hypersensitive response (HR) in *Arabidopsis*. The study showed that in *npr1 (GFP-ATG8a)*, *AvrRpt2* was more effective at inducing the production of autophagosome and autophagy flux than that in *GFP-ATG8a*. The mRNA expression of *ATG1*, *ATG6* and *ATG8a* were more in *npr1* during the early HR. Based on transcriptome data analysis, enhanced disease susceptibility 1 (EDS1) was up-regulated in wild-type (WT) but was not induced in *atg4a4b* (ATG4 deletion mutant) during *AvrRpt2* infection. Compared with WT, *atg4a4b* had higher expression of *salicylic acid glucosyltransferase 1 (SGT1)* and *isochorismate synthase 1 (ICS1)*; but less salicylic acid (SA) in normal condition and the same level of free SA during *AvrRpt2* infection. These results suggested that the consumption of free SA should be occurred in *atg4a4b*. *AvrRpt2* may trigger the activation of Toll/Interleukin-1 receptor (TIR)-nucleotide binding site (NB)-leucine rich repeat (LRR)—TIR-NB-LRR—to induce autophagy via EDS1, which was inhibited by nonexpressor of PR genes 1 (NPR1). Moreover, high expression of *NPR3* in *atg4a4b* may accelerate the degradation of NPR1 during *AvrRpt2* infection.

## 1. Introduction

Autophagy is a highly conserved intracellular degradation and recycling procession, which exists in yeast, plants and mammals. It controls cellular homeostasis, stress adaptation, and programmed cell death (PCD) in eukaryotes [1]. Autophagy acts as a key regulator of plant innate immunity and contributes with both pro-death and pro-survival functions to antimicrobial defenses [2]. ATG knockout mutants display impaired autophagy activity and fail to regulate hypersensitive response (HR)-PCD that initiated by *Psm* ES4326*/AvrRpt2 (AvrRpt2)* infection [3], which are recognized by *Arabidopsis* R proteins resistant to *Pseudomonas syringae* 2 (RPS2) [4]. The recognition by R proteins triggers signaling events of effector-triggered immunity (ETI), which is generally associated with immune responses [5]. Genetic screening identifies the signaling components of R protein mediated HR: The signals are generated by the binding of Toll/Interleukin-1 receptor (TIR) and leucine rich repeat (LRR) (TIR- nucleotide binding site (NB)-LRR) R protein. Then TIR-NB-LRR promoted enhanced disease susceptibility 1 (EDS1), which induced autophagy by resistant to *P. syringae* 4 (RPS4) under *AvrRps4* infection; non-race specific disease resistance (NDR1) is required for a different R proteins called coiled-coil (CC)-NB-LRR (CC-NB-LRR), which was mediated by RPS2 under *AvrRpt2* infection, and triggers HR independent of autophagy [6]. EDS1 was found to be a key mediator of autophagosome maturation, and without it, ATG4 would no longer be activated due to regulation from the ATG12–ATG5 complex, which has been described in previous work [7]. In this study, we explored autophagy induced by *AvrRpt2* during the initial of HR and the new functions of *ATG4*, related to *EDS1*.

Salicylic acid (SA) plays an important role in plant immunity against biotic and abiotic stress [8]. Previous work has suggested that plants synthesize SA from phenylalanine ammonia lyase (PAL) and isochorismate synthase 1 (ICS1) [9]. Once synthesized, SA undergoes a number of biologically relevant chemical modifications including glucosylation, methylation, and amino acid (AA) conjugation. Most modifications render SA inactive, while at the same time they allow fine-tuning of its accumulation, function and/or mobility [10]. Abiotic (e.g., UV-C) and biotic (e.g., *P. syringae*) stresses significantly induce the formation of free SA and SA glucose conjugates in *Arabidopsis*. Consistent with this induced response, SA glucosyltransferase (SGT) is induced by SA and appropriate biotic and abiotic stress, which catalyzes the conversion of SA into a conjugated form [11].

The components of the SA mediated immunity pathway are the nonexpressor of pathogenesis-related genes (NPR) proteins that include NPR1–NPR4 four close isoforms. NPR1 functions as a transcriptional activator, whereas NPR3 and NPR4 are transcriptional repressors. They all work independently and harmoniously to regulate the expression of downstream genes [12]. NPR1 is central to the activation of SA defense-related genes, such as PR genes [13], which is used as molecular markers for generating plant resistance responses [14]. Previous work suggested that plant autophagy operated a negative feedback loop modulating SA signaling to negatively regulate senescence and immunity-related PCD [15]. NPR3 and NPR4 function as adaptors of the Cullin 3 ubiquitin E3 ligase to mediate NPR1 degradation in an SA-regulated manner. NPR3 mediates NPR1 breakdown via 26S proteasome only in the presence of SA, while NPR4 does that only in its absence [16]. The roles of NPRs under *AvrRpt2* infection still need further study.

Here, we report that EDS1 is involved in the *AvrRpt2*-induced autophagy and that ATG4 inhibits the consumption of free SA and alleviates the degradation of NPR1, providing a new insight into the plant autophagy.

## 2. Results

### 2.1. NPR1 Inhibited AvrRpt2-Induced Autophagy

First, we checked the autophagic vesicle formation directly in the *GFP-ATG8a* and *npr1 (GFP- ATG8a)* with concanamycin A (ConA) and wortmannin (WM) by confocal microscopy under *AvrRpt2* infection (Figure 1A). Autophagic bodies were accumulated in the central vacuole of *GFP-ATG8a* cells upon *AvrRpt2* + ConA infection for 3 h. These puncta were more evident in equally treated *npr1 (GFP-ATG8a)* cells. Then we used 8.95 μM WM to block autophagy effectively in *Arabidopsis* [17]. As Figure 1A showed, WM blocked the formation of autophagic bodies in *GFP-ATG8a* and *npr1 (GFP-ATG8a)* under *AvrRpt2* infection as most of the fluorescence remaining diffuse within the cytosol (Figure 1A). This result suggested that the absence of NPR1 affect the formation of autophagosomes. Western blot technology was used to assess the release of autophagy flux (free GFP) and detect the degradation of *GFP-ATG8a* (Figure 1B). ATG8 proteins are lipidated with phosphatidylethanolamine (PE) to initiate autophagosome formation in autophagy process, and the outer membrane of the autophagosome subsequently fuses with the vacuolar membrane to transport the contents of the autophagic bodies into the vacuole, where GFP-ATG8a degraded to release a free, relatively stable GFP. Therefore, the levels of free GFP reflect the rate of autophagy [18]. The result showed that the level of free GFP in *GFP-ATG8a* and *npr1 (GFP-ATG8a)* increased with the time of *AvrRpt2* infection and both reached the maximum at 6 h, but decreased significantly at 12 h. The level of free GFP in each *npr1 (GFP-ATG8a)* group was higher than that in wild-type (WT) group (Figure 1B). These results showed that *AvrRpt2* induced the production of autophagosome and NPR1 inhibited the autophagy flux.

The core machinery of autophagy could be broken down into three functional units: ATG1-ATG13 comprising the kinase complex, an upstream regulator that initiates autophagosome formation; The ATG9 and ATG6/vps30 complexes are involved in vacuolar protein sorting and boosting phagophore expansion; Ubiquitin like conjugation systems (ATG5-ATG12 complex and ATG8-PE complex) are essential for autophagosome formation [19]. The expression of *ATG1*, *ATG6* and *ATG8a* in WT and *npr1* were examined by qRT-PCR (Figure 1C). Compared with WT, the result showed that the expression of *ATG1* increased at 3 h and then no longer changed in *npr1* with *AvrRpt2* treated for 9 h; the expression of *ATG6* in *npr1* increased at 3 h, decreased rapidly at 6 h, then gradually increased again at 9 h; the expression of *ATG8a* gradually increased until 6 h, then decreased at 9 h in *npr1*. In summary, these results suggested that NPR1 inhibits the mRNA expression of *ATG1*, *ATG6* and *ATG8*a during *AvrRpt2* infections.

### 2.2. EDS1 Was Up-Regulated Under AvrRpt2 Infection

ATG4 is a requisite factor in the ATG8 conjugation system. To investigate the autophagy induced by *AvrRpt2* and new function of ATG4, we performed transcriptome sequencing and analyzed on BMKCloud (www.biocloud.net) in WT and *atg4a4b* (ATG4 deletion mutant) under *AvrRpt2* infection for 12 h. The number R^2^, which represent that means the square of the Pearson coefficient, was larger than 0.85 for both the tested samples (Figure 2A). It demonstrated the experiment’s reliability and its usefulness in revealing differences in gene expression between samples. In total, 3518 and 2344 expressing genes were identified in the WT and *atg4a4b* under *AvrRpt2* infection, respectively. Among those, 977 genes were specific to the *atg4a4b*, whereas 2151 genes were specifically expressing in the control WT as demonstrated in the Venn diagram (Figure 2B). Kyoto Encyclopedia of Genes and Genomes (KEGG) pathway was used to reveal the differences in metabolic pathways. Each point in the Figure 2C represented a KEGG pathway. The path names were shown on the left axis. The abscissa was the enrichment factor, which represented the ratio of the proportion of genes differentially annotated to the pathway to the proportion of genes annotated to the pathway. The larger the enrichment factor, the more reliable the significance of the enrichment of differential genes in this pathway. According to the result, it had showed significant differences in the signal pathways of interaction between plants and pathogens, and plant hormone signal transduction when WT and *atg4a4b* infected with *AvrRpt2*. Then, the partly KEGG pathway was further analyzed. When WT was infected with *AvrRpt2*, its downstream factors salicylic acid glucosyltransferase 1 (*SGT1*) (Log_2_FC 1.6187) and heat shock protein 90 (*HSP90*) (Log_2_FC 1.0307) that associated with plant resistance showed red, which meant the expression was up-regulated. Surprisingly, the downstream factor *EDS1* (Log_2_FC 1.2281) in response to *AvrRps4* that triggered TIR-NB-LRR to induce autophagy also showed red. However, in *atg4a4b*, all the responding genes had no change under *AvrRpt2* infection (Figure 2D). In order to strengthen the conclusion, the mRNA expression of *EDS1* was checked in WT and *atg4a4b* (Figure 2E). The result showed that the expression of *EDS1* increased in WT but remained unchanged in *atg4a4b* under *AvrRpt2* infection for 12 h. The above results showed that EDS1 played a key role to autophagy induced by *AvrRpt2,* and ATG4 maybe had new functions of inhibiting the expression of *SGT1*, *HSP90* and *EDS1*.

### 2.3. ATG4 Inhibited the Occurrence of HR during AvrRpt2 Infection

To further study the function of ATG4 in *AvrRpt2*-induced autophagy, we did phenotypic experiments for intuitive exploration (Figure 3A). *rps2* was used as a negative control, which fails to recognize the AvrRpt2 effector [20]. At 1 d.p.i of *AvrRpt2* infection, *rps2* and *atg4a4b* had no obvious plaques, while the leaves of *npr1* and *atg8a* (ATG8a deletion mutant) shrunk. At 2 d.p.i of *AvrRpt2* infection, the leaves of *rps2* and *atg4a4b* were only yellowed, while the leaves of *npr1* and *atg8a* were especially degraded. Numbers are ratios of leaves with HR phenotype from by repeating the experiment independently (Appendix A). WT served as a positive control and *rps2* served as a negative control in ion leakage assay. The results showed that *npr1* had significantly higher ion concentration than WT and *rps2*; and *atg4a4b* had lower ion concentration than WT, but still higher than *rps2*; the trend of ion concentration in *atg8a* was consistent with WT (Figure 3B). Next, the expression of *pathogenesis-related 1 (PR1)* in *npr1*, *atg4a4b* and *atg8a* were examined (Figure 3C). The results showed that mRNA expression levels of *PR1* in *atg4a4b* and *atg8a* were higher in comparison to WT, indicating that ATG4 and ATG8a inhibited *PR1* mRNA expression. The above results indicated that ATG4 inhibited the occurrence of HR during *AvrRpt2* infection.

### 2.4. ATG4 Inhibited SA Consumption during AvrRpt2 Induced Autophagy-Dependent HR

We originally reported that SA-associated *EDS1* and *SGT1* expression was significantly up-regulated under *AvrRpt2* infection (Figure 2D). Firstly, contents of total SA and free SA in WT and *atg4a4b* with *AvrRpt2* infection for 12 h were tested. The results showed that the absence of ATG4 did not cause significant changes in SA content. After *AvrRpt2* infection, the total SA and free SA content in WT and *atg4a4b* were increased and the total SA of WT increased significantly more than that in *atg4a4b* (Figure 4A). ICS1 was the key enzyme for SA synthesis and these results prompted us to check the expression of *ICS1* to further study. The results in Figure 4B showed that the expression of *ICS1* had an increasing expression in WT and *atg4a4b* at the early stages of infection, which was more obviously in WT. After reaching the highest expression, both of them decline until 8 h. Then, WT declined slowly and finally reached stability, while *atg4a4b* continued to increase until it reached the highest expression level. NPR1, NPR3 and NPR4 are receptors for SA [12] and their gene expression was tested in WT and *atg4a4b*. The results of Figure 4C showed that the background expression level of *NPR4* in *atg4a4b* was higher than that in WT. The expression of *NPR1*, *NPR3* and *NPR4* in WT and *atg4a4b* increased, and then declined gradually with *AvrRpt2* infection time at the initial stage of infection. At 12 h, there was no difference in the expression of *NPRs* in WT and *atg4a4b*. *NPR3* showed the highest expression and *NPR4* was higher than NPR1. After 12 h, the expression of *NPR1*, *NPR3* and *NPR4* in *atg4a4b* rose again with the increase of infection time, then decline after reaching the highest expression level; while the expression of *NPR1* and *NPR3* in WT did not increase with the infection time prolonging, the expression of *NPR4* continued to increase with the infection time. In addition, NPR1 protein expression was tested by using NPR1 antibody. The NPR1 level was determined on the basis of the ratio of the NPR1 band intensity to that of the non-specific band (asterisk) [16]. We unexpectedly found that NPR1 protein was highly expressed in *atg4a4b*. With 12 h of *AvrRpt2* infection, the NPR1 protein expression level increased significantly in WT, *rps2*, *atg5* and *atg8a*, while decreased significantly in *atg4a4b* (Figure 4D). 

The mRNA expression of *ICS1* in *atg4a4b* increased, while total SA content decreased and free SA content unchanged when compared with WT at 12 h of *AvrRpt2* infection (Figure 4A,B), suggesting that ATG4 inhibited free SA consumption when responded to pathogen infection. Collectively, these results suggested that ATG4 inhibited the consumption of free SA, which promoted the interaction between NPR3 and NPR4 to accelerate the degradation of NPR1 during *AvrRpt2* induced autophagy-dependent HR.

## 3. Discussion 

Plants have evolved a multilayer immune system to recognize and respond to invading pathogens. The first layer includes pattern recognition receptors (PRRs) that detects conserved pathogen-associated molecular patterns (PAMPs) and initiates plant immune response by the name of PAMP-triggered immunity (PTI) [21]. Another layer uses resistance (R) proteins to directly or indirectly identify effectors of pathogen called ETI, then it triggers defense response rapidly that often includes a localized PCD reaction known as HR [22]. The most prevalent type of plant R proteins belongs to the NB-LRR class that can be further separated into TIR-NB-LRR associate with EDS1 and CC-NB-LRR R proteins [6].

*AvrRpt2* recognized by the CC-NB-LRR type R protein RPS2 and was considered to trigger hypersensitive cell death that was strictly NDR1-dependent but autophagy-independent [6,23]. For the first time, we found that *AvrRpt2* induced the production of autophagosomes and autophagy flux, which was inhibited by NPR1 (Figure 1A,B). In addition, NPR1 inhibited the mRNA expression of *ATG1*, *ATG6* and *ATG8a* in the early stages of HR (Figure 1C). These results indicated that the production of autophagosomes was induced by *AvrRpt2* and was suppressed by NPR1. The mRNA expression of other *ATG* genes in WT, such as the mRNA expression of *ATG4a* and *ATG4b* decreased and then rise; the mRNA expression of *ATG5* and *ATG12a* decreased with the infection time of *AvrRpt2* (data not shown); these are very interesting. Previous work showed that *AvrRpt2*-induced the expression of several primary jasmonic acid (JA)-responsive genes and JA is a positive regulator of RPS2-mediated ETI [24]. We speculated that *AvrRpt2*-induced autophagy maybe involve in other key hormone signals (for example JA) besides SA signal, which needed further study.

ATG4 is the only protease among dozens of ATG proteins. It also serves as a requisite factor in the ATG8 conjugation system: one of the unique mechanisms in autophagy [25,26]. Then, we further explored the mechanism with transcriptome analyses in WT and *atg4a4b*. We found that the expression of *EDS1*, *SGT1* and *HSP90* was up-regulated in WT, while there was no change in *atg4a4b* (Figure 2B,C). Similar result was showed in the mRNA expression of EDS1 with qRT-PCR (Figure 2E), suggesting that *atg4a4b* led to a concomitant EDS1 blockage. Therefore, the up-regulation of *EDS1* in WT was conducted by the R protein TIR-NB-LRR, which leaded to the occurrence of autophagy during *AvrRpt2* infection. 

Moreover, SGT1 and HSP90 as chaperone proteins are required for plant disease resistance and involved in SA metabolism [27,28,29]. Glucosylation inactivates SA and allows vacuolar storage of relatively large quantities of SA due to reduced toxicity. SGT1 catalyzes the conversion of free SA to salicylic acid 2-O-β-D-glucose (SAG) [11]. The results in Figure 2B,C showed that *SGT1* increased in WT, but remained unchanged in *atg4a4b.* The results in Figure 4b showed that the expression of *ICS1* in *atg4a4b* was higher than that in WT, indicating that the free SA in *atg4a4b* should be more than that in WT. Actually, total SA content in *atg4a4b* was lower than that in WT, while the free SA didn’t change between WT and *atg4a4b* after *AvrRpt2* infected for 12 h, we speculated that free SA in *atg4a4b* was consumed. In the early stage of HR induced by *AvrRpt2*, a large amount of SA was consumed in *atg4a4b*, resulting in a low concentration of free SA, which promoted the interaction of NPR3 and NPR1 [16]. The high mRNA expression of *NPR3* was detected in Figure 4C, leading the actual expression of NPR1 decreased after 12h infection. 

Therefore, these results showed that *AvrRpt2* induced autophagy also via EDS1 pathway. In the early stage of HR caused by *AvrRpt2*, ATG4 may suppress NPR3 synthesis via inhibiting the consumption of free SA and promote the expression of NPR1 (Figure 5). In the future, we will continue to explore the interaction between ATG4 and NPRs.

## 4. Materials and Methods 

### 4.1. Plant Materials and Chemical Treatment

Seeds were cultivated in soil culture in growth cabinets at 22 °C (day) and 18 °C (night), with a 16 h light period (120 μmol m^−2^ s^−1^) and 82% relative humidity for 2–4 weeks. The Arabidopsis thaliana mutants (in ecotype Col-0), *rps2*, ATG8a-lacking mutant (*atg8a*) and NPR1-lacking mutant (*npr1*) were provided by Dr. Xinnian Dong (Duke University, NC, USA). Transgenic *GFP-ATG8a* (Col-0 background) was donated by Dr. Kohki Yoshimoto of the Plant Science Center of Japan. *npr1* (*GFP-ATG8a*) was produced by crossing *npr1* and *GFP-ATG8a*. The *atg4a4b* was produced by crossing *atg4a* (SALK_085300) and *atg4b* (SALK_056994). *atg8a* (SALK_045344) was obtained from the *Arabidopsis* Biological Resource Center. SA was purchased from Sigma-Aldrich, China (S5922-100G; 239763-5GM-M, Shanghai, China). 

### 4.2. Pathogen Growth and Inoculation

The bacterial used in this study was *Psm* ES4326/*AvrRpt2* and was grown at 28 °C in King’s B medium containing 50 mg/L streptomycin and 10 mg/L tetracycline. Overnight log-phase cultures were collected by centrifugation, washed with 10 mM MgCl_2_, and then diluted to a final optical density of 0.02 at 600 nm (OD_600_).

### 4.3. SA Measurement

Following the previous procedure [30,31], 4-weeks-old plants leaves were used to measure the SA levels. High-performance liquid chromatography (HPLC) with fluorescence detectors (HPLC, Shimadzu LC-6A, Japan) was used to analyze total extracted SA and free SA, at a 294 nm excitation wavelength and a 426 nm emission wavelength.

### 4.4. Total RNA Extraction and Quantitative Reverse Transcription-PCR (qRT-PCR)

Total RNAs were extracted from *Arabidopsis* leaves at indicated times after the treatment of TRI reagent according to the manufacturer’s instruction (Invitrogen, Carlsbad, CA, USA). The first-strand complementary DNA was synthesized from total RNA using a Reverse-iT first-strand synthesis kit (Perfect Real Time, RR047Q, TaKaRa, Dalian, China). qPCR was performed using the ChamQ SYBR qPCR Master Mix (Low ROX Premixed, Q331-02/03, Vazyme Biotech Co., Ltd., Nanjing, China) on ABI Life QuantStudio 6. The thermal cycles of qPCR were initial denaturation at 95 °C for 30 s, followed by 40 cycles by 40 cycles at 95 °C for 5 s and 60 °C for 34 s. *Ubiquitin 5* and *AtACTIN2* were used as internal control. The primers used are listed in Appendix A.

### 4.5. Protein Extraction

Amounts of 0.4 g leaves were ground in liquid nitrogen and the powder was resuspended in 1 mL ice-cold extraction buffer comprised of 50 mM Tris-HCl (pH 7.5), 150 mM NaCl, 5 mM EDTA, 0.2% (*w*/*v*) Triton X-100, 0.2% (*w*/*v*) Nonidet P-40 and 1% (*w*/*v*) phenylmethanesulfonyl fluoride (PMSF). To extract NPR1, add 40 µM MG115, 1% β-ME, 500× protease inhibitor cocktail (PMSF not included) and 5000× phosphatase inhibitor cocktail to the protein extraction buffer. The extracts were centrifuged and the protein concentration of the supernatant was determined with a Bio-Rad protein assay.

### 4.6. Western Blotting

Total proteins were extracted from leaves at the indicated time points after different treatments with the protocol of Karppinen [32]. The antibodies used for western blotting included actin (Engibody Biotechnology, AT0004, WB: 1:3000, Dover, DE 19901, USA) and GFP (JL-8 Monoclonal Antibody, A-6455, WB: 1:5000, Fisher, Invitrogen, Waltham, MA, USA). Detection was performed using a LI-COR Odyssey Infrared Imaging System (LI-COR, Inc., Lincoln, NE, USA). NPR1 protein expression was tested by using NPR1 antibody. The NPR1 level was determined on the basis of the ratio of the NPR1 band intensity to that of the non-specific band (asterisk) [16].

### 4.7. Confocal Microscopy

*GFP-ATG8a* and *npr1 (GFP-ATG8a)* seedlings (7 days old) were treated with MgCl_2_ or 1 μm concanamycin A (ConA) (Invitrogen, Waltham, MA, USA) or ConA + *Psm* ES4326*/AvrRpt2* or ConA + 8.95 μm wortmannin (WM) (19545-26-7, MCE, NJ, USA) + *Psm* ES4326*/AvrRpt2* for 3 h. Root epidermal cells below the cotyledon were imaged using a Zeiss LSM880 confocal laser scanning microscope. GFP fluorescence was excited by a 488 nm argon laser and detected at 505–550 nm by a photomultiplier detector. At least 10 sets of images were obtained for quantification analysis. GFP was counted per section according to 10 sets of images field of vision.

### 4.8. Ion Leakage

Ion leakage assay was performed as previously described [33]. The leaves of 4-week-old WT, *rps2*, *npr1*, *atg4a4b* and *atg8a* plants were infiltrated with *Psm* ES4326/*AvrRpt2*, and 6 leaf discs (8 mm diameter) were removed rapidly following infection and washed in 50 mL ddH_2_O (twice). After 10 min, we removed the wash water and replaced it with 15 mL of ddH_2_O. Ion leakage was then measured over time.

### 4.9. Transcriptome Analysis

WT and *atg4a4b* were infected by *Psm* ES4326*/AvrRpt2* for 12 h, and then their samples were collected by taking 8–10 real leaves from three different culture pots as three biological replicates. Total RNA was extracted by TRIzol Reagent (Invitrogen, Thermo Fisher Scientific, Shanghai, China). RNA quality was determined using Agilent 2100 Bioanalyzer (Agilent Technologies Canada Inc., Mississauga, ON, Canada). RNA libraries were constructed from 2 lg of total RNA and subjected to deep sequencing at an Illumina Hiseq 2500 platform (BioMarker Technologies Illumina, Inc., Shanghai, China).

## Figures and Tables

**Figure 1 ijms-21-05147-f001:**
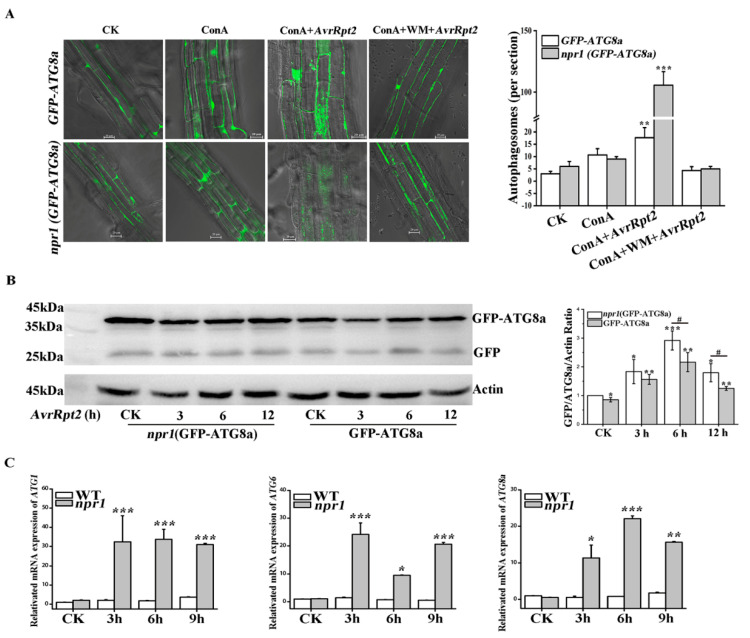
Nonexpressor of PR genes 1 (NPR1) inhibited *Psm* ES4326*/AvrRpt2 (AvrRpt2)*-induced autophagy. (**A**) Autophagosomes formation. *GFP-ATG8a* and *npr1 (GFP-ATG8a)* treated with four groups: MgCl_2_; concanamycin A (ConA); ConA + *AvrRpt2* and ConA + wortmannin (WM) + *AvrRpt2* and then examined by confocal microscopy. Scale bars, 20 µm. Numbers of puncta per section in the root cells of *GFP-ATG8a* or *npr1 (GFP-ATG8a)* seedlings in the left. *n* = 10 sections from three independent experiments per genotype. (**B**) Western Blot to detect autophagy flow in *GFP-ATG8a* and *npr1 (GFP-ATG8a)* when plants treated with *AvrRpt2* at 3 h, 6 h and 12 h and quantitative analyses of GFP/GFP-ATG8a/Actin ratio. Each data is three independent replicates. Each value is the mean ± SD of three replicates. Statistically significant differences between treatments (# *p* < 0.05, * *p* < 0.05, ** *p* < 0.01 and *** *p* < 0.001). (**C**) Quantitative RT-PCR data showed the expression of *ATG1*, *ATG6* and *ATG8a* in wild-type (WT) and *npr1* after *AvrRpt2* infiltration for 3 h, 6 h and 9 h. The CK group was treated with MgCl_2_ as control. Each data is three independent replicates. Each value is the mean ± SD of three replicates. Statistically significant differences between treatments (* *p* < 0.05, ** *p* < 0.01 and *** *p* < 0.001).

**Figure 2 ijms-21-05147-f002:**
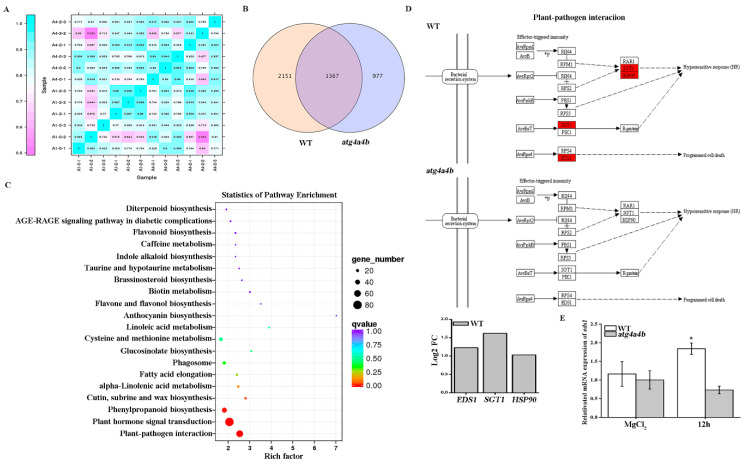
Enhanced disease susceptibility 1 (EDS1) was up-regulated under *AvrRpt2* infection. (**A**) Correlation between RNA-Seq samples. A 1-0-1/2/3 represent three replicates of WT, A 1-2-1/2/3 represent three replicates of WT infected with *AvrRpt2*, A 4-0-1/2/3 represent three replicates of *atg4a4b* and A 4-2-1/2/3 represent three replicates of *atg4a4b* infected with *AvrRpt2*, heat maps of the correlation coefficient between samples, the number represent R^2^ that means the square of the Pearson coefficient. (**B**) Venn diagram of expressed genes in WT and *atg4a4b* when infected with *AvrRpt2*. FPKM > 1 is the expression threshold. (**C**) Transcriptome analysis of Kyoto Encyclopedia of Genes and Genomes (KEGG) enrichment in WT and *atg4a4b* when infected with *AvrRpt2* for 12 h. (**D**) Transcriptome analysis of KEGG pathway of plants and pathogens interaction. The red color means that the expression was up-regulated. (**E**) The mRNA expression of *EDS1* in WT and *atg4a4b*. Each data is three independent replicates. Each value is the mean ± SD of three replicates. Statistically significant differences between treatments (* *p* < 0.05).

**Figure 3 ijms-21-05147-f003:**
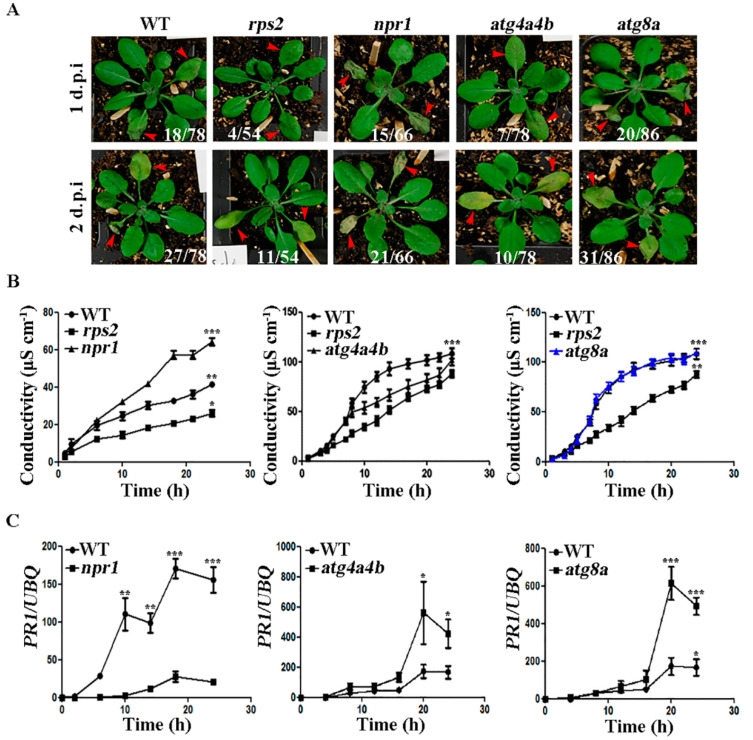
ATG4 inhibited the occurrence of hypersensitive response (HR) during *AvrRpt2* infection. (**A**) Phenotypes of WT, *rps2, npr1, atg4a4b* and *atg8a* after *AvrRpt2* infiltration for 1 or 2 days. Numbers are ratios of leaves with HR phenotype. (**B**) Ion leakage assay in WT, *rps2*, *npr1*, *atg4a4b* and *atg8a* when infected with *AvrRpt2*. (**C**) Quantitative RT-PCR data showed the expression of pathogenesis-related *(PR1)* in WT, *npr1*, *atg4a4b* and *atg8a* when infected with *AvrRpt2*. Each value is the mean ± SD of three replicates. Statistically significant differences between treatments (* *p* < 0.05, ** *p* < 0.01 and *** *p* < 0.001).

**Figure 4 ijms-21-05147-f004:**
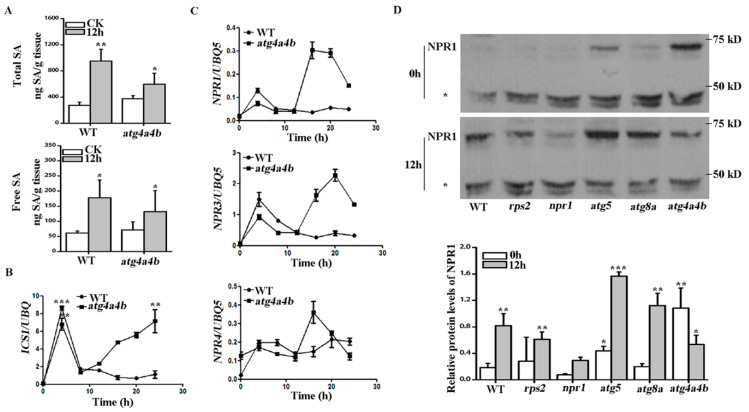
ATG4 inhibited salicylic acid (SA) consumption during *AvrRpt2* induced autophagy-dependent HR. (**A**) Total SA and free SA contents in WT and *atg4a4b* under *AvrRpt2* infection for 12 h. (**B**) Quantitative RT-PCR data showed the expression of *isochorismate synthase 1 (ICS1)* in WT and *atg4a4b* when infected with *AvrRpt2*. (**C**) Quantitative RT-PCR data showed the expression of *NPR1, NPR3 and NPR4* in WT and *atg4a4b* when infected with *AvrRpt2*. (**D**) Western Blot to detect NPR1 in WT, *rps2, npr1, atg5, atg8a* and *atg4a4b* when plants treated with *AvrRpt2* for 12 h and quantitative analyses of the results of NPR1 by statistical methods. The NPR1 level was determined on the basis of the ratio of the NPR1 band intensity to that of the non-specific band (asterisk). Each value is the mean ± SD of three replicates. Statistically significant differences between treatments (* *p* < 0.05, ** *p* < 0.01 and *** *p* < 0.001).

**Figure 5 ijms-21-05147-f005:**
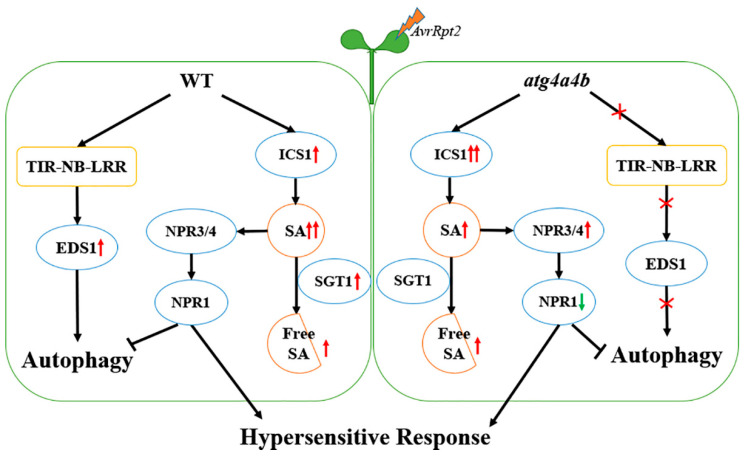
Working model. *AvrRpt2* induced autophagy-dependent hypersensitive response via EDS1. ATG4 may suppress NPR3 synthesis via inhibiting the consumption of free SA and promote the expression of NPR1. The red arrow means up-regulated and the green arrow means down-regulated. Two arrows indicate high degree.

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
