# Peer review of "ATG4 Mediated Psm ES4326/AvrRpt2-Induced Autophagy Dependent on Salicylic Acid in Arabidopsis Thaliana"

_ijms, 2020, doi:10.3390/ijms21145147_

Round 1

Reviewer 1 Report

The paper “ATG4 Mediated Psm ES4326/AvrRpt2-induced Autophagy Dependent on Salicylic Acid in Arabidopsis thaliana” by Gong and colleagues presents observations suggesting that AvRpt2 induces autophagy and that NPR1 and the salicylic acid signaling pathway may be involved in the control of autophagy in these conditions.

I have 3 main comments:

  • Most of the experiments/results are interesting, yet without any indication of biological repeatability it is difficult to take them for more than mere observations. The authors should provide the results of at least 3 independent biological repeats for all experiments and give more information on the statistical tests that were used to demonstrate the significance of the results.
  • It is very hard to follow the authors rationale throughout the paper and a great effort should be put towards rewriting the manuscript to explain how and why the authors perform their experiments, the biological meaning of the results, how they fit with current models and what novelty they bring to the understanding of plant immunity. For example, the introduction is difficult to follow for non-specialists with lots of very long names of genes and pathways and no clear connection between them: a schematic would be very helpful.
  • In my opinion, for the most part, the authors conclusions are not fully supported by the current data and additional experiments need to be performed.

  1. NPR1 inhibited AvrRpt2-induced autophagy

I am not quite convinced that AvrRpt2 induces autophagy and the results need to be significantly improved with additional biological experiments and statistical analyses to show significance in differences between the mutant and WT.

Fig.1 A- The scale bar is missing and the quality of the images is very poor, we can observe a very large number of punctate structures within the cells but I would find it hard to quantify those precisely. The fluorescent puncta also look fairly small compared to what is observed in other papers or in my own personal experience. Are they autophagosomes? The authors should use wortmannin to block autophagosome formation and validate that the puncta are genuine autophagy structures. Further, the authors need to add concanamycine A to conclude whether the absence of NPR1 induces the formation of novel autophagosomes or prevents their degradation (ie, assays autophagy flux).

Additionally, in the right panel (quantification) the authors should provide more information about the unit of ‘autophagosomes’ (number of autophagosomes per what?) and need to say how many independent biological experiments they have performed.

Fig. 1B – Need quantification of 3 independent experiments, especially given that the differences between conditions are not striking, we need to assess the significance of such differences. It also surprising that the number of autophagosomes in Fig 1A is very high but that the degradation of GFP-ATG8 (autophagy flux) appears to be rather low. It questions even further the nature of the puncta presented in Fig.1A. Can the authors comment on that? I do not think that the band right below GFP-ATG8 corresponds to GFP-ATG8 PE, is that a urea gel? if so the authors should be more specific in their methods section. Without proper control (ie, a mutant where ATG8 lipidation is prevented), the authors cannot conclude about the nature of this band, it might as well be a non-specific band of any kind.

Fig. 1C – I would not say that ATG2, ATG9 or ATG18 are part of the ATG1 activation complex, please rephrase. The figure needs additional results of independent biological experiment not only replicates. From the results, it looks like treatment with the AvrRpt2 has no effect on the expression of ATG genes even though autophagy is supposed to be induced. Why would the npr1 mutant have an induced expression of those genes? The authors need to comment/discuss.  

  1. EDS1 was upregulated under AvrRpt2 infection

For the remaining part of the manuscript, the authors went on to analyze the difference between WT plants and the atg4a atg4b mutant where autophagy is completely abolished. In this whole part, the text is misleading in the fact that it suggests that ATG4 is specifically or directly involved in some SA-mediated mechanism when all the phenotypes could (and likely) result from blocking autophagy rather than deleting ATG4 per se. This should be tested by looking at another mutant where autophagy is completely prevented. For example, the authors write that: ‘The deletion of ATG4 failed to up-regulate EDS1 to activate TIR-NB-LRR under AvrRpt2 infection, which eventually disable the induction of autophagy’. Suggesting that this is the reason why there is no autophagy in the atg4 mutant when actually, there is no autophagy because ATG4 is critical for ATG8 lipidation and indispensable for AP formation.  

Fig. 2. I do not have major comments regarding the experiments but I disagree with the authors conclusion: ‘The above results showed that EDS1 played a key role to autophagy induced by AvrRpt2 (line 121)’. No, it does not. The results show that upon AvrRpt2 infection, there is an increased expression of EDS1 and that blocking autophagy prevents such an induction. If the authors want to explore the function of EDS1, they need to measure the autophagy flux in an EDS1 knock-out mutant compared to WT plants in these conditions.

Similarly, the authors write in the discussion: ‘Therefore, after AvrRpt2 infection, the up-regulation of EDS1 regulated the R protein TIR-NB-LRR, which lead to the occurrence of autophagy’ (line 212). There are no data presented here supporting that conclusion, this is only a model that the authors propose. Further, the authors should discuss why the deletion of ATG4 failed to upregulate EDS1? Any proposed mechanism?

  1. ATG4 inhibited the occurrence of HR

Fig. 3, the authors observe that the leaves of the npr1 and atg8a mutant are shranked: what does that mean biologically? Also, the atg8a mutant is not supposed to exhibit large phenotypes as autophagy is largely compensated by other ATG8 isoforms. These experiments need much further explanation: what do the results mean? For non-specialists it is not easy to follow. Why did the authors select these different mutants, for instance: why rps2? Is that some kind of a control? What is the expected phenotype for this mutant? Fig. 3A needs to be more compelling: with proper quantification of the phenotypes with statistical analyses of at least 3 independent experiments.

  1. ATG4 inhibited SA consumption during AvrRpt2 induced autophagy-dependent HR

In this part, the authors conclude that “ATG4 inhibited the consumption of free SA, which promoted the interaction between NPR3 and NPR1 to accelerate the degradation of NPR1 in atg4a4b at 12 h after AvrRpt2 infection.” (L. 179-182).

None of these conclusions are backed up by the data.

(1) ‘ATG4 inhibited SA consumption’:  the authors show a slight decrease in total SA in the atg4 mutant compare to WT upon infection. First, there are no statistical analyses showing that this difference is significant, the authors should provide additional independent experiments (at least 3, with proper statistics). Second, the authors show that ICS1 is upregulated in these conditions (Fig.4B). Given that the protein encoded by this gene is involved in SA production, and that there is, supposedly, a little bit less SA in the atg4 mutant, the authors conclude that ATG4 inhibits SA consumption. I disagree with this conclusion, an increase in gene expression does not systematically translates into an increase in protein level (which is balanced-out by both synthesis and degradation) and even if it did, it does not mean that the product of the protein will increase! What if another factor is limiting? Based on the results presented here, the authors cannot conclude anything about SA consumption.  

(2) ‘which promoted the interaction between NPR3 and NPR1 to accelerate the degradation of NPR1 in atg4a4b at 12 h after AvrRpt2 infection’ (line 180-182). The authors did not provide any data regarding NPR3 and NPR1 interaction, where does that come from? Additionally, I have a lot of trouble with the results regarding NPR1 degradation presented in Fig. 4D. First, is that an antibody against NPR1? It is not mentioned in the methods section. Please provide the reference. Second, I am not convinced about the specificity of this antibody for NPR1? Why is there still a band in the npr1 mutant? Third, the quantification should be made with a housekeeping protein, not an unspecific ban which might as well be a product of degradation of NPR1 for all we know. Fourth, please provide the number of independent biological experiments that were made of this experiment. Fifth, why don’t the authors mention the results with the atg5 and the atg8a mutants? How do they fit in the model? In the discussion, the authors add ‘these consumed SA may interact with NPR3 and accelerate the degradation of NPR1 during AvrRpt2 infection’. Yet, is there a degradation of NPR1 during AvRPT2 infection? Looking at Fig.4D (and assuming that the band is actually NPR1) it actually looks like the level of the protein is increasing upon AvRpt2 infection.

Reviewer 2 Report

This manuscript assumes a connection between autophagy and salicylate under a certain setup which involves a pathosystem rather well-established and commonly used among relevant community members. The aims are clearly defined and the introduction, the discussion has a short format making the manuscript appealing, without the usual "heavy" tedious format.

However, from a textual/content perspective, the manuscript is impossible to read and needs serious linguistic improvements as I find hard to get the conveyed messages and comprehend the biological links suggested. I am sure there are quite many "cryptic messages" under the veil of textual misuses spread evenly throughout the main body of the manuscript. The insights that can be considered novel are rather limited.

Some interpretations from the authors (and I guess by myself due to difficulties to understand parts of the manuscript) are rather misleading, such as for example, the blockage of ATG4 that would lead to a concomitant EDS1 blockage and this will reduce autophagy. I think authors here may have attempted to build concepts on positive feedback loops imposed by autophagy (which are consistent with their overall time-scale and experimental design; see further below though on the timings). Mechanistic links to the suggested processes are missing and some of them are rather far-fetched, circumstantial and not supported by the available data. Overall, I reckon this work as rather confirmatory and an incremental addition in a large list of many others connecting autophagy and salicylate (I recommend authors properly cite previous works).

Some further comments are listed below: 

Abstract: I think needs to be re-written. Impossible to follow.

Line 60: what is really the new insight here?

Line 76: please correct spelling

Figure 1: The micrographs are not convincing. Where are the autophagosomes? The imaging plane is not the same among samples. I do not see the changes implied either in intensity or puncta numbers. Selection of different focal plane is sort of "cherry-picking" and authors should be careful as in WT the section is midplane while in the mutant is peripheral. Furthermore, it is unclear  Actin is saturated and the loading cannot be assumed.

Figure 4: SA changes here do not support the hypotheses on different SA levels. The variances here are limiting potential effects. Furthermore, the induction of npr1 would be anyways limited at 20h due perhaps to the increased occurrence of cell death in the atg4 mutant.

Figure 5: The model is impossible to follow and contradicts itself. ATG4 depletion would anyways deplete autophagy irrespective of EDS1. I do not understand how they fit in the same pathway. Furthermore, the SA context is not easy to perceive. 

What would be the rate of autophagy in npr1? This would support/refute the suggested model.

Reviewer 3 Report

Very interesting work that complements the understanding of the interaction of the plant host - pathogen.

Round 2

Reviewer 2 Report

in this version, the authors took into consideration some of my comments. However, the manuscript is not much improved and still linguistically suffers. Especially, the microscopy still is not adequate (atg8)
